# Trends in Adopting Industry 4.0 for Asset Life Cycle Management for Sustainability: A Keyword Co-Occurrence Network Review and Analysis

**Sachini Weerasekara, Zhenyuan Lu, Burcu Ozek, Jacqueline Isaacs and Sagar Kamarthi \***

Department of Mechanical and Industrial Engineering, Northeastern University, Boston, MA 02115, USA
\* Correspondence: s.kamarthi@northeastern.edu

**Abstract:** With the potential of Industry 4.0 technologies to enable sustainable manufacturing, asset life cycle management (ALCM) has been gaining increasing attention in recent years. This study explores the evolution of Industry 4.0 technology applications to sustainable ALCM from 2002 to 2021. This study is based on keywords collected from 3896 ALCM-related scientific articles published in the Web of Science, IEEE Xplore and Engineering Village between 2002 and 2021. We conducted a review analysis of these keywords using a network science-based methodology, which unlike the tedious traditional literature review methods, gives the capability to analyze a huge number of scientific articles efficiently. We built keyword co-occurrence networks (KCNs) from the keywords and explored the network characteristics to uncover meaningful knowledge patterns, knowledge components, knowledge structure, and research trends in the body of literature at the intersection of ALCM and Industry 4.0. The network modeling and data analysis results identify the emerging Industry 4.0-related keywords in ALCM literature and indicate the recent explosion of connectivity among keywords. We found IoT, predictive maintenance and big data to be the top three most popular Industry 4.0-related keywords in ALCM literature. Furthermore, this study maps relevant ALCM keywords in contemporary literature to the nine pillars of Industry 4.0 to help the responsible manufacturing community identify research trends and emerging technologies for sustainability.

**Keywords:** asset life cycle management (ALCM); sustainability; Industry 4.0; keyword co-occurrence network (KCN); literature review

## 1. Introduction

Industry 4.0 refers to advanced technologies that bring connectivity, intelligence, agility and digitalization to industrial applications. Many consider it a disruptive technological wave that will significantly impact industries, society and the environment [1]. Industry 4.0 technologies facilitate collecting, sharing, and analyzing real-time data to connect the cyber world and the physical world. This convergence of cyber and physical worlds creates digital production systems that are decentralized, flexible, and resource efficient; they can manufacture individualized products on demand [2]. In addition, technological advances, such as big data and predictive analytics, pave the way for minimizing waste and resource consumption [3].

Asset life cycle management (ALCM) in manufacturing is increasingly adopting Industry 4.0 technologies for achieving sustainability goals. "Asset" refers to any piece of property owned by a person or a firm. The asset life cycle comprises everything that occurs from the identification of the need for the asset until its disposal [4]. Traditionally, asset management has been identified as "a strategic, integrated set of comprehensive processes (financial, management, engineering, operating, and maintenance) to improve lifetime effectiveness, utilization and return from physical assets (production and operating equipment and structures)" [5]. Typically, asset management approaches focus primarily on cost minimization. However, in the current industrial setup, a strong positive correlation

exists between environmental, social, governance (ESG) ratings and the return on assets [6]. The idea that organizations can achieve long-term benefits by focusing on sustainable development has motivated manufacturing firms to adopt Industry 4.0 technologies for ALCM. Asset-related decision making holds the utmost importance in ALCM [7,8]; this is where decision support tools offered by Industry 4.0 mainly come into play.

The generally recognized nine pillars of Industry 4.0: advanced simulation, autonomous robots, system integration, additive manufacturing, big data, augmented reality, Internet of things (IoT), cloud computing and cybersecurity [9]. At the foundation of Industry 4.0 is cyber–physical systems (CPS), in which physical objects and infrastructure are controlled by computer-based algorithms that integrate sensing, networking and computation. CPS provides the foundation for the Internet of things (IoT), which provides smart objects with embedded sensors, software and other technologies to connect with other objects. Such connected things fetch and exchange data through the Internet [8,10] and in networks of devices [11]. Big data analytics is used to access vast amounts of data and provide fast decision making by processing and learning from the data [12,13]. Big data complements IoT to enable decision making. Another Industry 4.0 tool is advanced simulation, which supports the prediction of stochastic processes occurring in the physical world [14]. Following Table 1 in [15], Table 1 presents some recent studies that focus on applying the nine pillars of Industry 4.0 to ALCM, for sustainability in the manufacturing sector. The small sample of studies presented in Table 1 puts forward IoT, big data and cloud computing as highly popular Industry 4.0 pillars used in the ALCM literature, and this indication motivated us to carry out an analysis on a larger sample of studies.

**Table 1.** Publications that discuss applications of Industry 4.0 for ALCM for sustainability.

| Article/s | Contents | Industry 4.0 Pillar/s |
|---|---|---|
| [16–19] | Performance prediction and long-term system behavior monitoring using digital twins (DT); digital continuity along different life cycle phases of the system, to improve maintenance decision making. | Advanced Simulation |
| [20–23] | Equipment and product energy consumption and emission monitoring and management. | IOT, Big Data, Cloud Computing |
| [24,25] | Tracking life cycle data using RFID to improve end of life processing. | IOT |
| [26–28] | Implementation of IOT technologies for improving re-manufacturing efficiency. | IOT |
| [29–31] | Deep learning and big data for learning complex system behavior to predict future states; optimizing decision-making throughout the entire life cycle using real time data and information to achieve improvements in energy savings and fault diagnosis. | Big Data, Cloud Computing |
| [32,33] | Energy efficient machining optimization through CPS and big data. | Autonomous Robots, Big Data, Cloud Computing |
| [34,35] | Machine conditioning monitoring system using plant automation technologies. | IOT, Autonomous Robots, Cloud Computing |
| [36–38] | Predictive maintenance of assets using machine learning and chronicle mining. | Big Data, Cloud Computing |

Despite the increasing attention on Industry 4.0-enabled ALCM for sustainability, research trends in applying Industry 4.0 for sustainability in ALCM are rarely explored. In this paper, we conducted a review of the existing knowledge structure, knowledge components, and research trends concerning Industry 4.0 applications in ALCM for sustainability.

We conducted a network science-based literature review of 3896 ALCM-related scientific articles published from 2002 to 2021. A conventional literature review is inherently tedious, time consuming and not conducive to processing content from a large number of

papers. Hence, we used the keyword co-occurrence network (KCN) methodology [39,40] to identify the knowledge structure and knowledge components of Industry 4.0 applications in ALCM. KCN models each keyword as a node and the co-presence of two keywords in a publication as an edge connecting the respective nodes. The number of times each keyword pair co-occurs in the body of literature is captured by the weight of the edge connecting the keyword pair. We implemented this KCN methodology on the keywords collected from 3896 ALCM-related articles to explore the critical network characteristics. A statistical and visual analysis of the KCN results were performed to study the evolution of the Industry 4.0 applications in ALCM for sustainability.

The remainder of this article is organized as follows. The Methodology section introduces the KCN methodology and describes the data collection and cleaning process. The Results section provides an analysis of the results of the experiments. The Discussion section examines the existing knowledge structure and insightful trends observed in the results. The Conclusions section summarizes the findings and states future research directions.

## 2. Methodology

This section provides a detailed description of the KCN methodology, the data collection and processing workflow, and key network parameters for drawing insights into the characteristics of Industry 4.0 applications in ALCM.

### 2.1. The KCN Methodology

Citation network and KCN are two network-based methods that have been widely used for analyzing the contents of scientific articles. A citation network identifies popular studies based on their citation frequencies. It studies the scientific information transmission in the domain by focusing on the associations among the cited work [41,42]. However, it does not reveal emerging trends in the literature. In contrast, a KCN explores the keywords in scientific articles to understand the knowledge components [43,44]. Keywords give a quick overview of the prominent elements of the scientific article [45]. A KCN captures associations among different knowledge components and measures the relative importance of each component within the network. For the current study, the KCN methodology is more appropriate than using citation network methods because we are interested in analyzing the knowledge components of the articles rather than in identifying the importance and popularity of articles (studies). This paper implements KCNs to analyze the evolution of the topics in the scientific studies on Industry 4.0 applications in ALCM over a period of two decades.

### 2.2. Data Collection and Processing

We collected literature related to applications of Industry 4.0 in ALCM for sustainability from three databases: Web of Science, IEEE Xplore Digital Library and Engineering Village. These databases provide access to more than 37,000 scientific articles. We used the following query to collect articles published between 2002 and 2021 in the three databases.

> ('sustainable') AND ('asset life cycle') AND ('industry 4.0' OR 'cyber physical' OR 'big data' OR 'robotics' OR 'IOT' OR 'IIOT' OR 'augmented reality' OR 'additive manufacturing' OR 'cloud' OR 'system integration' OR 'simulation' OR 'cybersecurity') AND ('manufacturing')

Articles were partitioned into four 4-year windows: 2002–2006, 2007–2011, 2012–2016, and 2017–2021. A separate KCN was created for each window to investigate the evolution of topics on Industry 4.0 for ALCM over a period of two decades. Figure 1 presents the data collection and pre-processing procedure. The initial collection included 5907 articles: 3613 articles from Engineering Village, 1801 articles from Web of Science, and 493 articles from IEEE Xplore. After cleaning the initial set of records by removing records with missing data and duplicates, 3896 articles were input to the next step, i.e., keyword extraction. We used natural language processing (NLP) methods to extract and process the keywords

from the scientific articles. Then we processed these keywords to eliminate redundancy by unifying hyphenated and non-hyphenated phrases (e.g., cyber-physical system and cyber physical system), singular and plural variants (e.g., "decision support system" and "decision support systems"), synonyms (e.g., "performance measurement" and "performance monitoring") and acronyms (e.g., "IOT" and "internet of things"). Keyword processing yielded 212, 632, 1885, and 3519 keywords for the periods 2002–2006, 2007–2011, 2012–2016 and 2017–2021, respectively. A separate adjacency matrix was built for each of the above four windows using the co-occurrence counts of all the pairs of keywords in all the papers belonging to each time window. Adjacency matrices were then input to different functions in the Python Networkx package and other custom-built Python functions to obtain network parameters and drive insights into the knowledge structure.

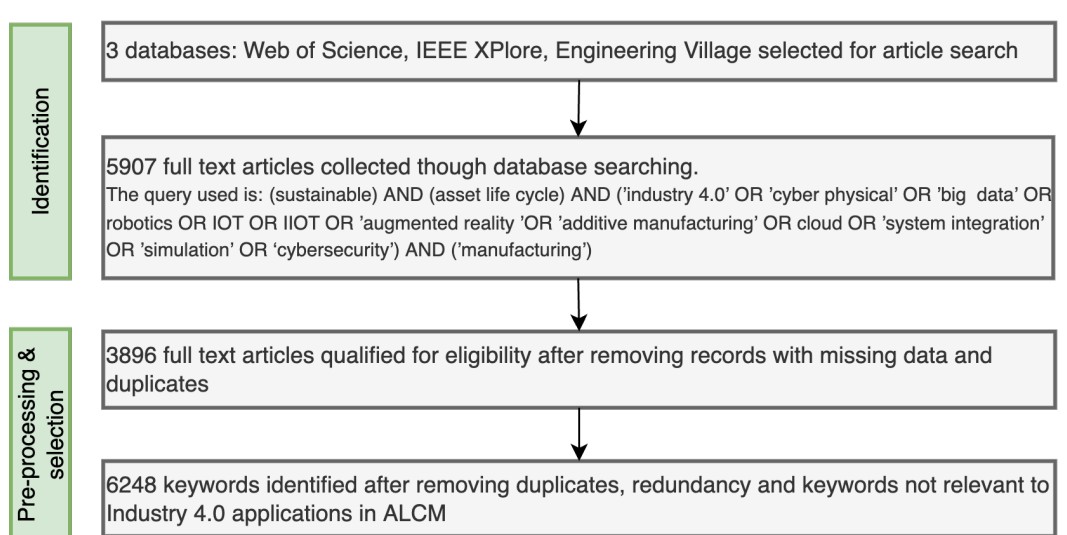

**Figure 1.** Data collection and pre-processing procedure.

### 2.3. Network Analysis Parameters

This section introduces key parameters [39,46,47] used in the current study for analyzing the topology of the ALCM-keyword networks.

The *degree* ($k_i$) of a node is the total number of edges connected to it. The *degree* of a node (keyword) represents the number of different nodes (keywords) with which it has synergy. The *degree* of a node is a measure of its association with other nodes.

$$k_i = \sum_{j \in \mathbf{V}} e_{ij}$$

where $\mathbf{V}$ is the set of nodes in the network. $e_{ij} \in \{0, 1\}$, where $e_{ij} = 1$ if an edge exists between node $i$ and node $j$, and $e_{ij} = 0$ if otherwise. The higher the degree, the higher the node's importance in the network.

Degree values are normalized by dividing them by the maximum possible degree, i.e., $N - 1$, where $N$ is the total number of nodes in the network, to obtain the *degree centrality* [46] ($dc_i$), which is a key measure of the relative popularity of a keyword within the network.

$$dc_i = \frac{\sum_{j \in \mathbf{V}} e_{ij}}{N - 1}$$

The weight ($w_{ij}$) of an edge $e_{ij}$ is the number of times node $i$ and node $j$ co-occur. Weights of all the edges connected to node $i$ are summed to obtain the *strength* ($s_i$) of node $i$, which is an indicator of the popularity of node $i$ in the network.

$$s_i = \sum_{j \in \mathbf{V}} e_{ij} w_{ij}$$

Another important parameter is the *average weight as a function of end point degrees*:

$$< w_{ij} > \sim (k_i k_j)$$

where $k_i$ and $k_j$ denote degrees of nodes $i$ and $j$, respectively, $k_i k_j$ is the end point degree of edge $(i, j)$, and $w_{ij}$ is the weight of the edge $(i, j)$. The average weight $< w_{ij} >$ is the average of the weights of all edges whose end point degree is equal to $k_i k_j$. The average weight as a function of end point degrees is useful to understand the strength of the association between nodes of different degrees. If $w_{ij}$ proportionally increases with $k_i k_j$, it is an indication that *connections among the high-degree keywords are more than the connections among the low-degree keywords*. If $w_{ij}$ decreases with $k_i k_j$, it is an indication that *connections among the low-degree keywords are more than the connections among the high-degree keywords*.

*Average weighted nearest neighbor's degree $k_i^w$* of a node is another key parameter of a KCN.

$$k_i^w = \frac{1}{s_i} \sum_{j \in \mathbf{V}} e_{ij} w_{ij} k_j$$

It gauges the likelihood that nodes connect with neighboring nodes having similar characteristics, i.e., high-degree nodes frequently connecting with high-degree neighboring nodes and low-degree nodes frequently connecting with low-degree neighboring nodes. If high-degree keywords show high values or low-degree keywords show low values for this measure, it is an indication that high-degree keywords mostly associate with high-degree keywords and low-degree keywords associate with low-degree keywords. Otherwise, it can be inferred that high-degree and low-degree keywords associate with both high-degree and low-degree keywords equally.

*Weighted clustering coefficient* ($c_i^w$) of a node is the final parameter explored in this work. It is a measure of *a node's cohesiveness with its neighbors*. It accounts for the local structure clustered around a node in terms of the interaction intensity found in the local triplets [47]. A high $c_i^w$ value reveals the strong cohesion among the keywords centered around a keyword of interest.

$$c_i^w = \frac{1}{s_i(k_i - 1)} \sum_{j,h \in \mathbf{V}} \frac{w_{ij} + w_{ih}}{2} e_{ij} e_{ih} e_{jh}$$

## 3. Results

Table 2 presents the topological properties of KCNs in the four time windows. As can be seen in Figure 2a, over the period from 2002 to 2021, the number of scientific articles on Industry 4.0 adoption for sustainable ALCM increased by approximately 12 fold, indicating a significant expansion of the body of knowledge over the years. The number of keywords (nodes) increased by approximately a factor of 26, and the total number of connections among keywords (edges) increased by a factor of 75; this indicates the emergence of many novel keywords, the explosion of connectivity among these keywords, and the emergence and convergence of many new research topics.

From Figure 2b, it is clear that the average degree and the average strength of keywords rose at an increasing rate, hinting at a rapidly growing synergy among topics and concepts about sustainable ALCM literature. While average values of degree, strength, and weight have continued to increase over the years, maximum values of degree, strength, weight have increased at a much higher rate compared to the average values. This indicates the formation of strong and distinct keyword hubs with the expansion of a number of keywords and connections. Maximum degree, strength, and weight significantly increased from from 2011 onward, indicating that the earlier part of the second decade (between 2012

and 2016) may have given rise to many novel keywords that became topics of significant interest in the subsequent years. Figure 3a,b also illustrates that while the median of node degree and strength have slightly increased over time, a considerable number of dominant nodes with high degree and strength have emerged with time in the literature. Figure 3c shows an interesting change in the pattern of weights. In the 2002–2006 period, weights follow a heavily right-skewed lognormal distribution. However in the later time windows, a majority of the weights have became almost uni-valued and a small number of high-valued weights have come into existence. This implies that a few keyword pairs have become the focus of researchers.

**Table 2.** Topological properties of four KCNs built for the four time windows.

| Metric | 2002–2006 | 2007–2011 | 2012–2016 | 2017–2021 |
|---|---|---|---|---|
| No. of articles | 181 | 506 | 1085 | 2124 |
| No. of nodes | 123 | 619 | 1607 | 3160 |
| No. of edges | 320 | 2066 | 8465 | 24,162 |
| Av. degree | 5.2 | 6.67 | 10.53 | 15.29 |
| Max. degree | 21 | 44 | 73 | 247 |
| Av. strength | 6.76 | 8.4 | 12.34 | 18.24 |
| Max. strength | 26 | 61 | 168 | 497 |
| Av. weight | 1.3 | 1.25 | 1.17 | 1.19 |
| Max. weight | 4 | 6 | 12 | 40 |

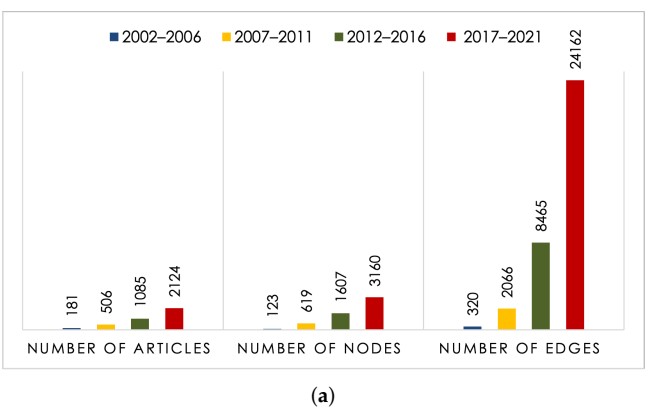

(a)

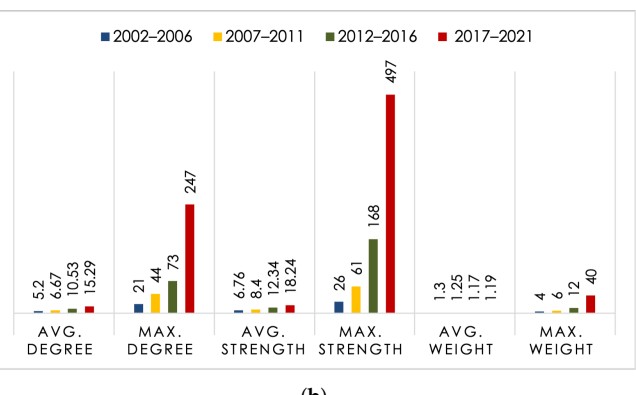

(b)

**Figure 2.** Barplots of KCN topology properties across the four time windows. (**a**) Trends in number of articles, nodes and edges. (**b**) Trends in node degree, node strength, and edge weights.

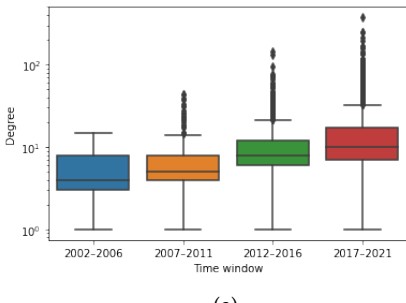

(a)

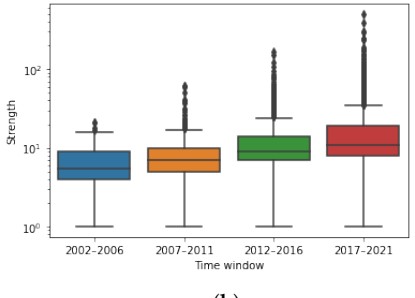

(b)

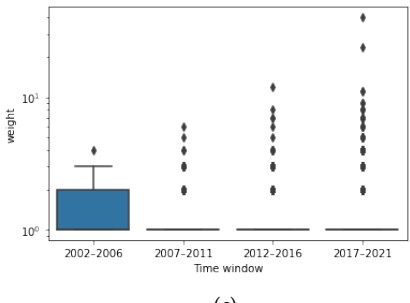

(c)

**Figure 3.** Boxplots of node degree, node strength and edge weight across the four time windows. (**a**) Boxplots of node degree. (**b**) Boxplots of node strength. (**c**) Boxplots of edge weights.

Table 3 presents the first 20 keywords sorted in descending order of degree ($k_i$) in each KCN time window. It shows the evolution of the popularity of dominant keywords in the network and identifies keywords that are gaining popularity in recent years.

From 2002–2006 to 2007–2011, degree values increase, the degree of centrality decreases, and new keywords appear in the list of top 20 keywords. This indicates a growth in the volume of keywords, loss of popularity of the keywords that may have been established in the period 2002–2006 and the emergence of novel keywords during 2007–2011. The emergence of new keywords continued during 2012–2016, and these newly created keywords remained active in the 2017–2021 period. It can be seen that Industry 4.0 related terms, such as IoT, big data and artificial intelligence, have gained importance in ALCM literature from 2012 onward and continued to increase in popularity.

**Table 3.** Top 20 keywords with the highest degree ($k_i$) and degree centrality ($dc_i$) values.

| 2002–2006 | | 2007–2011 | | 2012–2016 | | 2017–2021 | |
|---|---|---|---|---|---|---|---|
| **Keyword** | $k_i, dc_i$ | **Keyword** | $k_i, dc_i$ | **Keyword** | $k_i, dc_i$ | **Keyword** | $k_i, dc_i$ |
| Digital business | 21, 0.172 | Resource management | 44, 0.0890 | Decision making process | 73, 0.059 | IOT | 247, 0.122 |
| Web based management system | 21, 0.172 | Knowledge management | 39, 0.066 | Knowledge management | 72, 0.053 | Industry 4.0 | 166, 0.078 |
| Business process | 18, 0.147 | Decision support system | 37, 0.061 | Life cycle management | 46, 0.043 | Big data | 136, 0.058 |
| Project management | 17, 0.139 | Asset management | 37, 0.061 | Decision support system | 45, 0.037 | Artificial intelligence | 117, 0.047 |
| Business model | 17, 0.139 | Sustainable development | 31, 0.051 | Continuous improvement | 45, 0.036 | Circular economy | 115, 0.047 |
| Global enterprise sustainability | 17, 0.139 | Operational efficiency | 26, 0.038 | IOT | 40, 0.030 | Energy efficiency | 111, 0.039 |
| ICT | 16, 0.139 | Continuous improvement | 23, 0.037 | Asset integration | 35, 0.028 | Decision making process | 110, 0.037 |
| Decision support system | 15, 0.126 | Multi criteria analysis | 21, 0.032 | Cyber-security | 34, 0.026 | Digital | 108, 0.037 |
| knowledge share | 14, 0.114 | Smart meter | 21, 0.032 | Performance measurement | 34, 0.026 | Life cycle | 108, 0.037 |
| Internet | 12, 0.106 | Life cycle assessment | 16, 0.027 | Big data | 33, 0.024 | Integration | 98, 0.035 |
| Resource management | 12, 0.106 | Real time energy management | 14, 0.021 | Maintenance | 33, 0.024 | Predictive maintenance | 94, 0.034 |
| Asset management | 11, 0.106 | Data | 9, 0.016 | Resilience | 32, 0.024 | Decision support system | 93, 0.034 |
| Multi agent based simulation | 11, 0.106 | Real time decision making | 9, 0.016 | Technology innovation | 31, 0.023 | Information and communication | 80, 0.031 |
| Intranet | 9, 0.081 | Equipment reliability | 9, 0.016 | ICT | 27, 0.020 | Digital twin | 78, 0.031 |
| Open-source software | 8, 0.073 | Return on assets | 8, 0.014 | Energy efficiency | 26, 0.019 | Blockchain | 64, 0.028 |
| Generalized asset optimization | 7, 0.073 | Particle swarm optimization | 8, 0.014 | Cyber-physical systems | 23, 0.018 | Cyber-physical systems | 61, 0.0278 |
| Life cycle cost | 6, 0.065 | Technology innovation | 7, 0.014 | Communication | 22, 0.018 | Machine learning | 52, 0.024 |
| Machine fault diagnosis | 5, 0.059 | Cloud computing | 7, 0.014 | Smart meter | 22, 0.018 | Cloud computing | 52, 0.023 |
| Environmental policy | 4, 0.040 | Information technology | 6, 0.012 | Real time monitoring | 18, 0.016 | Data analysis | 46, 0.019 |
| Business process management | 3, 0.031 | Waste recovery | 6, 0.012 | Data security | 18, 0.016 | Digital asset | 45, 0.019 |

Figure 4 presents the rise or decline of keywords in their popularity in the time period from 2012 to 2021, using two slope charts. The number next to each keyword is its rank in the KCN after sorting them in descending order of strength, i.e., a rank of one (1) is given to the keyword with the highest strength. The slope charts demonstrate the shift in the ranks of keywords from 2012–2016 to 2017–2021. The slope chart on the left shows emerging keywords, which improved their ranks in 2017–2021, while the slope chart on the right gives declining keywords, which have lost their popularity in 2017–2021. Considering the slopes of the dotted lines connecting the ranks of the keywords in the two time windows, we can categorize the keywords into three groups. If the slope of the dotted line is almost zero, we can say that it is a primary topic that has been favored during both of the time windows. If the slope is positive, it is an emerging topic; if the slope is negative, it is a declining topic.

- *Primary topics*: Decision support system seems to be a favored topic over the past decade, and it is continuing to gain popularity. This can be referred to as a primary topic that has shaped ALCM literature.

- *Emerging topics*: Topics, such as machine learning, artificial intelligence, predictive maintenance, and data analysis, show a significant increase in their popularity in the latest time window compared to their presence in the previous years. In addition, IoT, cloud computing, and big data have continued to gain increasing popularity.
- *Declining topics*: Topics outside Industry 4.0, such as decision making, simulation, and system dynamics, show a declining pattern.

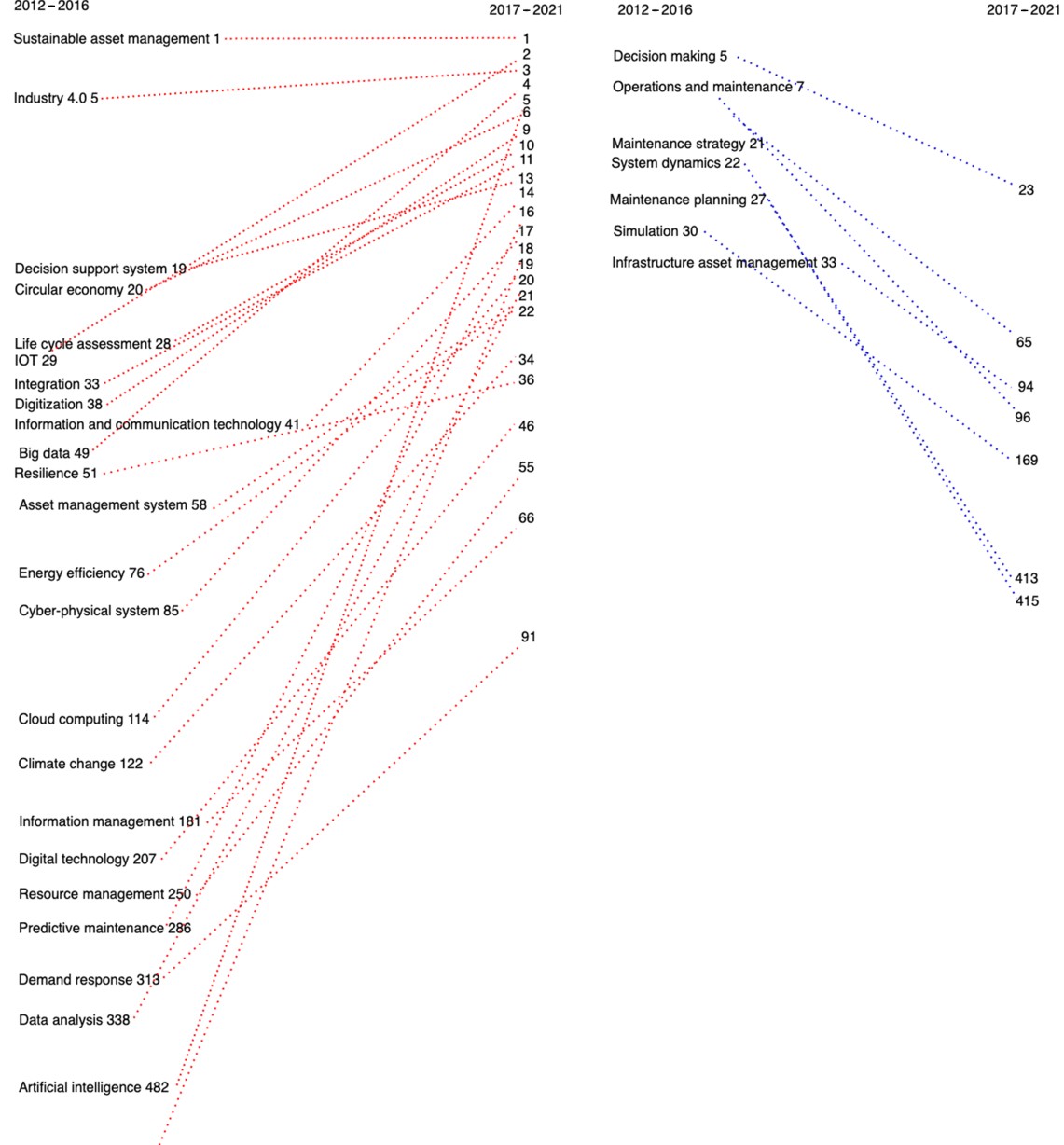

**Figure 4.** Slope charts of emerging keywords (**left**) and declining keywords (**right**). The value next to each keyword represents the strength-based ranking of the keyword in the time window.

Table 4 presents keywords with the highest positive differences in strength based rankings between 2012–2016 and 2017–2021, i.e., a rank of one (1) is given to the most popular keyword. High positive difference in the strength based ranking indicates a rise in the popularity of the keyword in 2017–2021. All of the keywords shown in the table have gained considerable increase in popularity in 2017–2021, with machine learning, artificial intelligence, data analysis, predictive maintenance and demand response improving by more than 200 ranks.

**Table 4.** Top 10 emerging Industry 4.0-related keywords, sorted in descending order of the difference of ranks.

| Keyword | SBR 2012–2016 | SBR 2017–2021 | Rank Diff. |
|---|---|---|---|
| Machine learning | 590 | 19 | 571 |
| Artificial intelligence | 482 | 5 | 477 |
| Data analysis | 338 | 22 | 316 |
| Predictive maintenance | 286 | 17 | 269 |
| Demand response | 318 | 91 | 227 |
| Cloud computing | 114 | 20 | 95 |
| Cyber-physical system | 85 | 18 | 68 |
| Big data | 49 | 4 | 45 |
| Digitization | 38 | 9 | 29 |
| IOT | 29 | 2 | 27 |

Figure 5 depicts four important network parameters of the KCNs for the four time windows: 2002–2006, 2007–2011, 2012–2016 and 2017–2021. Note that both x-axes and y-axes of these charts are on a logarithmic scale. Figure 5a presents probability distribution functions (PDF) of edge weights of KCNs built for each period. All four PDFs show a decaying pattern, indicating that heavy-weight edges are found less frequently than light-weight edges in all four KCNs. In other words, the number of light-weight edges is significantly larger than the number of heavy-weight edges. This observation suggests that Industry 4.0 in sustainable ALCM literature has a few popular keywords that go together with many other keywords, and the majority of the keywords co-exist with only a few other keywords.

Figure 5b presents the average weight as a function of endpoint degrees, which measures the impact of end node degrees of an edge on the weight of the edge. For all four time windows, average edge weights increase with endpoint degrees, which implies that keyword hubs are connected by heavy-weight edges. This association means that a small set of prominent keywords (hubs) co-occur in multiple articles. Average weight values sharply increase for $k_i k_j > 10^3$, indicating a strong positive correlation between the degree of endpoint nodes and the frequency of co-occurrence of keywords. For all four periods, most of the average weights are close to one, indicating the presence of many weakly connected keywords and a few heavily connected keyword hubs.

Figure 5c presents the trend in the average weighted nearest-neighbor degree as a function of the node degree. This observation indicates that, for other than 2002–2006, $k_i^w$ remained constant or slightly increased with the node degree. This result implies that high-degree keywords equally co-occur with both high-degree and low-degree keywords.

Figure 5d shows how the weighted clustering coefficient changes with the degree. The decreasing pattern indicates that the local structure of high-degree nodes in KCNs are much less clustered than that of low-degree nodes. Clustering measures how well the neighborhood is connected, where high clustering indicates a highly synergistic keyword group. The decline in clustering coefficients indicates that high-degree keywords are connected to neighboring keywords that do not form cohesive structures among them. However, low-degree keywords are connected to neighboring keywords with tightly connected local structures. It points to the tendency of high-degree keywords to connect with a wide variety of keywords and low-degree keywords with well-connected keyword groups. This behavior implies the emergence of a knowledge structure around a few select keyword hubs. In general, the trend line for 2017–2021 is below those of other time windows. This indicates that in the most recent time window, the new keywords are connected with less clustered neighbors. However, the fluctuations at high-degree nodes in the 2027–2021 period could be due to the dynamically changing synergy between popular keywords and other recently added keywords.

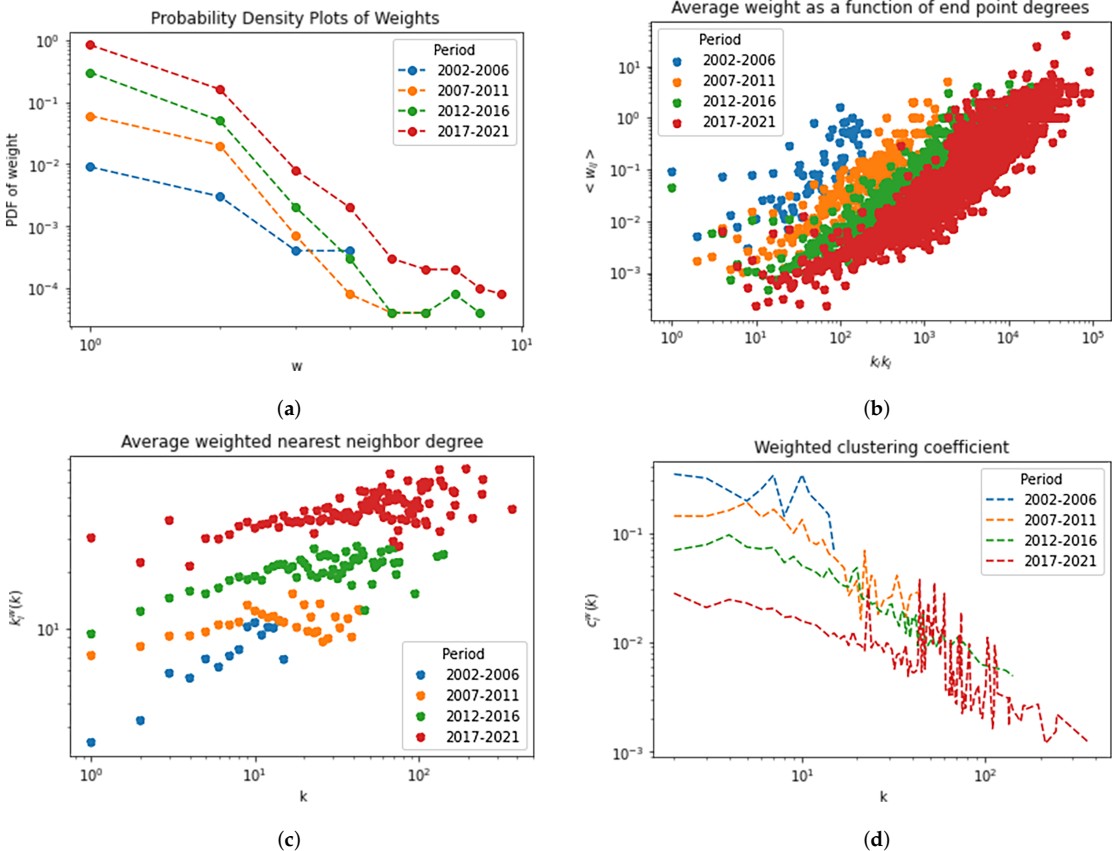

**Figure 5.** Key KCN matrices for 2002–2006, 2007–2011, 2012–2016 and 2017–2021. (**a**) Weight distribution. (**b**) Average weight vs. end point degree. (**c**) Average weighted nearest neighbor degree vs. degree. (**d**) Weighted clustering coefficient vs. degree.

Table 5 presents ALCM keywords with the highest connection intensity to their neighbors in the KCN of 2017–2021, and their Industry 4.0-related neighboring keywords. ALCM topics, such as intelligent asset, data-driven circular economy, asset life cycle management, master data management and asset maintenance management, frequently associate the given Industry 4.0-related neighboring keywords, while keywords such as IoT and big data are commonly associated with most of the ALCM topics presented in Table 5.

**Table 5.** Top 5 ALCM keywords with the highest weighted clustering coefficients, i.e., having highest connection intensities with their neighbors, and Industry 4.0-related neighboring keywords.

| ALCM Keyword | Clustering Coefficient | Industry 4.0-Related Neighboring Keywords |
|---|---|---|
| Intelligent asset | 0.038 | {Industry 4.0, Cloud, IoT, Crowdsense, Sensor data} |
| Data-driven circular economy | 0.03 | {IoT, Intelligent asset, Big data, Crowdsense} |
| Asset life cycle management | 0.0219 | {Big data, Industry 4.0, IoT, Predictive maintenance, Artificial intelligence} |
| Master data management | 0.019 | {Big data, IoT, Semantic web technology} |
| Asset maintenance management | 0.015 | {IoT, Statistical learning method, Sensor data, Big data, Artificial intelligence, Neural networks, Data mining, Predictive maintenance} |

## 4. Discussion

From 2002 to 2021, the number of studies on Industry 4.0 applications in ALCM increased 12 fold, while the number of keywords and connections between keywords increased 26 fold and 75 fold, respectively. The volume of the literature grew considerably, and so did the number of keywords, indicating the emergence of novel topics and broadening of the knowledge base. As the number of new keywords increased, the associations between them have grown three times faster. From 2012 onward, new keywords emerged from the research articles covering digital twin, IoT, and big data applications on ALCM, which experienced a rapid expansion. With the number of keywords increasing by 26 fold and connections by 75 fold, Industry 4.0-enabled sustainable ALCM has experienced a widening and deepening knowledge structure in recent years.

Although the size of KCNs expanded over the years, the average weight of KCNs did not vary much over the four time periods. This observation indicates that the average connection density of the KCNs remained the same over the years, although the distribution of the connections to nodes became uneven. The left-skewed distribution of node degrees and decreasing weight clustering coefficients imply that most keywords are low degree, and most of the edges are light weight. In other words, the KCNs have only a few keywords that are prominently connected with other keywords in the field. This trend might be because many researchers published scientific articles on popular topics.

A few keywords (e.g., IoT, big data, and Industry 4.0) co-occur with a wide variety of other keywords, while most of the keywords co-occur with only a few other keywords. High-degree keywords (e.g., IoT) tend to co-occur with both high-degree and low-degree keywords. While lesser explored topics appear in a small number of keyword groups, well-known keywords appear in many keyword groups. This shows that the high-frequency keywords are associated with many different subtopics of ALCM. Many keywords have lost popularity over time, and new keywords have been emerging for the past decade. IoT, big data, artificial intelligence, machine learning, and predictive maintenance are some emerging keywords, with artificial intelligence and machine learning showing considerable growth in the last decade. Table 6 presents the top 10 ALCM-Industry 4.0 keyword pairs in the ALCM literature that are connected by heavy-weight edges, i.e., most frequently co-occurring keywords, in the 2017–2021 KCN. Note that we included only ALCM-Industry 4.0 keyword pairs in Table 6, ignoring keyword pairs not relevant to the ALCM-Industry 4.0 nexus. It can be seen that IoT co-occurred the most with asset management. In addition, keywords such as predictive maintenance, decision support systems and big data show high co-occurrence frequencies with ALCM keywords. These trends reveal future research and technology directions.

**Table 6.** Top 10 most frequently co-occurring ALCM-Industry 4.0 keyword pairs in the ALCM KCN of 2017–2021.

| ALCM-Industry 4.0 Keyword Pair | Weight of the Edge Connecting the Pair ($w_{ij}$) |
|---|---|
| Asset management—IoT | 16 |
| Asset management—Industry 4.0 | 15 |
| Demand response—Industry 4.0 | 13 |
| Asset management—Predictive maintenance | 12 |
| Sustainability—IoT | 11 |
| Smart asset—IoT | 9 |
| Asset management—Big data | 7 |
| Supply chain—IoT | 6 |
| Circular economy—IoT | 6 |
| Life cycle—Predictive maintenance | 6 |

Technologies that have been gaining popularity over the last four years can be categorized as data analytics (data mining and machine learning), cloud-based technologies (big data, and IoT), and cyber–physical technologies (digital twins, and cybersecurity).

With the dawn of the smart manufacturing era, the life cycle management of manufacturing assets has become technology driven. As a result, the research focus has been shifting to cyber technologies that enable efficient and sustainable management of manufacturing assets. These developments are expected to continue to grow in the near future. KCN topology properties for the four time windows indicate that the co-occurrence of keywords has grown significantly over the years. However, the cohesiveness of most keywords weakened from the earlier years to the later years, as many new weakly connected keywords have emerged in recent years. This indicates that researchers, in recent years, have been exploring innovative approaches to life cycle management of manufacturing assets leveraging Industry 4.0 technologies. Overall, the research community of Industry 4.0 for sustainable ALCM, IoT, and other cloud-based technologies has proliferated, and the knowledge structure is expanding with a greater convergence.

Figure 6 presents the authors' subjective mapping of ALCM keywords relevant to the nine pillars of Industry 4.0. Popular Industry 4.0-associated keywords found in the KCN for 2017–2021 are given on the left, and the nine pillars of Industry 4.0 are given on the right. According to the authors' subjective mapping, cloud computing, IoT, and big data are the three most popular pillars with the highest representation in the KCN, which indicates that they play a significant role in defining the current technology trends in sustainable ALCM.

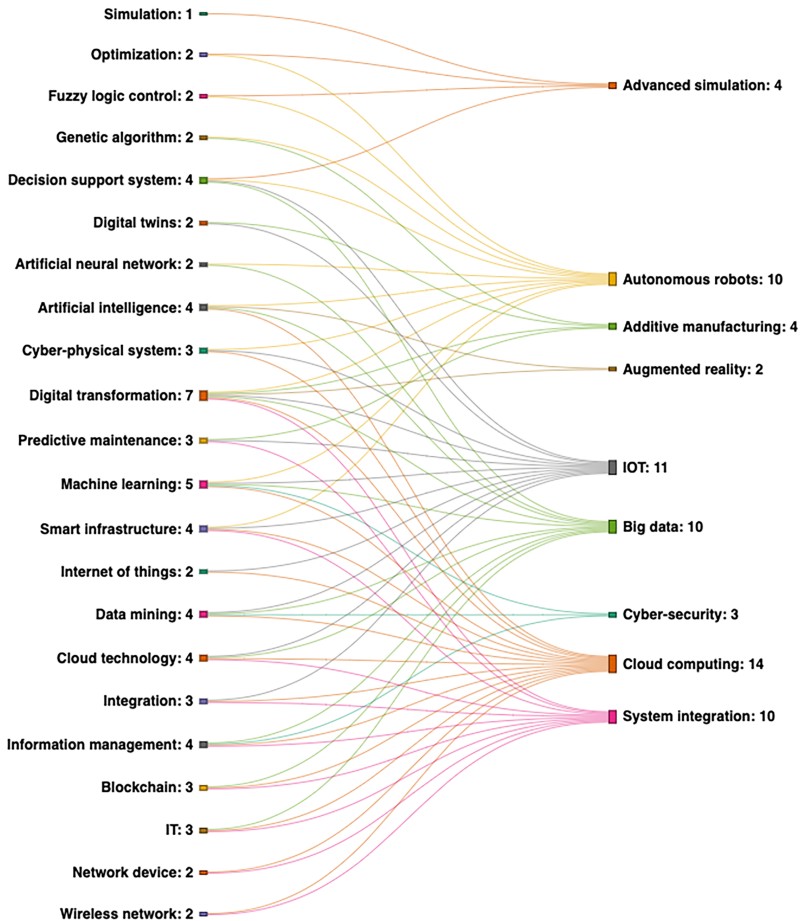

**Figure 6.** The authors' subjective mapping of top 22 Industry 4.0 keywords for 2017–2021 (**left**) ranked in the order of their strength to the nine pillars of Industry 4.0 (**right**). The value next to each keyword indicates the count of Industry 4.0 pillars associated with the keyword. Similarly, the number next to each pillar indicates the number of keywords associated with the pillar.

## 5. Conclusions

This study performed a keyword co-occurrence network (KCN) analysis of keywords in 3896 scientific articles related to Industry 4.0 as applied to ALCM for sustainability. These articles were published during the period from 2002 to 2021. The analysis reveals the evolution of knowledge structure over the past 20 years and the current and expected trends in the literature. The statistical analysis of network characteristics provides insights into trends in ALCM research (e.g., big data, artificial intelligence, IoT, and cloud computing) over time. Traditional literature reviews discuss methodologies and experimental findings. In contrast, KCN provides a macro-level understanding of the evolution of the knowledge structure and knowledge components to inform researchers about the declining and emerging research topics in the literature. It summarizes key characteristics of the body of literature and enables researchers to understand the big picture of knowledge trends quickly.

This study maps high-frequency ALCM keywords to nine pillars of Industry 4.0: advanced simulation, system integration, autonomous robots, augmented reality, additive manufacturing, Internet of things (IoT), big data, cloud computing, and cybersecurity. The popularity of each pillar provides insights into future research directions. Results depict that, currently, the top three most popular pillars in sustainable ALCM research are big data, IoT, and cloud computing. The KCN-based review and analysis results presented in this paper can serve as a road map for conducting a rigorous systematic review of the literature on Industry 4.0 technologies for ALCM.

Although this analysis uses only keywords to build the KCN and is as objective as possible, it may still have bias if the authors failed to identify vital terms as keywords. Due to the limitations of the natural language processing methods, some distorted, irrelevant, and redundant keywords might have made their way in to the final keyword list used for building KCNs. However, the effect of such noise in the keyword list is not likely to alter the observations made in this work. The KCN-based approach is otherwise very effective in reviewing the knowledge structure and research trends macroscopically. Future work could include extracting keywords from the articles' titles to make the knowledge coverage more comprehensive.

**Author Contributions:** Conceptualization: S.K., S.W. and J.I. Methodology: S.K., Z.L., B.O. and S.W. Software: Z.L. Formal analysis: S.W., Z.L. and B.O. Validation: B.O., Z.L. and S.W. Investigation: S.W., Z.L., B.O., S.W. and J.I. Resources: J.I. Data curation: S.W., Z.L. and B.O. Writing—original draft preparation: S.W. Writing—review and editing: S.K. and J.I. Supervision: J.I. and S.K. All authors have read and agreed to the published version of the manuscript.

**Funding:** This research received no external funding.

**Institutional Review Board Statement:** Not applicable.

**Informed Consent Statement:** Not applicable.

**Data Availability Statement:** Not applicable.

**Conflicts of Interest:** The authors declare no conflict of interest.

## Abbreviations

The following abbreviations are used in this manuscript:

| | |
|---|---|
| ALCM | Asset Life Cycle Management |
| KCN | Keyword Co-occurrence Network |
| IoT | Internet of Things |
| DT | Digital Twin |
| RFID | Radio-frequency Identification |
| CPS | Cyber–Physical Systems |
| KPI | Key Performance Index |
| TBL | Triple Bottom Line |

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
