# Peer review of "Trends in Adopting Industry 4.0 for Asset Life Cycle Management for Sustainability: A Keyword Co-Occurrence Network Review and Analysis"

_sustainability, doi:10.3390/su141912233_

Round 1

Reviewer 1 Report

The main question addressed by the authors was to explore the evolution of Industry 4.0 technology applications to sustainable Asset Life Cycle Management over the period from 2002 to 2021. It is relevant for industry 4.0 researchers for the manufacturing community to identify research trends and emerging technologies for sustainability. The novel approach of keyword co-occurrence network (KCN) was applied to address the trends in adopting industry 4.0 technologies. It fills a gap in the existing publications for manufacturing community to identify research trends and emerging technologies for sustainability.
It is well written with some minor corrections.
Conclusions are consistent with the evidences and argument are well presented like “Although this analysis using only keywords to build the KCN is as objective as possible, it may still have bias, if authors failed to identify vital terms as keywords. Due to the limitations of the natural language processing methods, some distorted, irrelevant and redundant keywords might have made their way in to the final keyword list used for building KCNs.”

Study was conducted for the period from 2002-2021, however I don't see any reference for the year 2021 which may be added. 

Author Response

Please find attached authors' response to Reviewer 1 comments.

Reviewer 2 Report

Dear Authors,

I would like to congratulate you on your work. Here are a few comments to help you improve your contribution.

Before the title I think you have to change the type of the paper from article to review

Abstract

Line 7- 8.- the term “knowledge components” is repeated.

At the end of the abstract, the results of the research should be described

Introduction

Line 48 & 53.- You speak about the environmental, social and economic aspects of sustainability, this concept is known as the triple bottom line, therefore, it would be great that you could introduce this term in the introduction.

Methodology

Data collection: you used the words cyber physical in your query, but you don’t talk about cyber physical systems when you speak about Industry 4.0 and their technologies. I suggest you introduce this topic in the introduction section

The keywords identified in the KCN should be seen before the Network Analysis Parameters showing a first result of the keyword extraction and process.

 Results

The results obtained are very general and I can’t easily see the relationship between industry 4.0 and LCM, which is the title of the paper. It would be necessary to analyze the results deeper, including examples and particular keywords, not only the global results and numbers.

Figure 4 should be modified or transformed into a table because it is not easy to understand, maybe you should use more colors or some graphs

Author Response

Please find attached authors' response to Reviewer 2 comments.

Reviewer 3 Report

1- Abstract

Your abstract should be improved for better reading and understanding. It is important to remember that the abstract is intended to convince the reader that the text is worth reading. Therefore, the implications of the analyses carried out and the novelty of the research in the area should be highlighted.

Line 8 – Delete repeated word “knowledge components”

2- Introduction

Table 1- What is the conclusion from this table?

Line 49 - Where are the links between ALCM and the nine Industry 4.0 pillars for economic, social, and environmental sustainability in the manufacturing sector?

Table 1 - It is advised that authors take into account Table 1 from the following studies, which has a comparable outcome shown in the table.

- Kuzior, A., & Sira, M. 2022. A Bibliometric Analysis of Blockchain Technology Research Using VOSviewer. Sustainability, 14(13), 8206.

3- Materials and Methods

Line 72 – Propose replace "materials and methods" with "methodology."

4- Results

Figure 4 shows the ranking and popularity of every keyword, but how were they all sorted into the three categories of primary, emerging, and declining topics?

5- References

This paper reviewed 3896 scientific articles published between 2002 and 2021. Most of the references, though, are outdated. Zero articles in 2021 and just one from 2020. References should contain more recent ones, particularly those from the previous three years.

Line 434 - A framework for Big Data driven product life cycle management. Is this a book or an article? - Update the formatting if possible.

Author Response

Please find attached authors' response to Reviewer 3 comments.

Round 2

Reviewer 2 Report

All the suggestions have been considered so I think that this paper is ready to be accepted